# OPTIMAL TRANSPORT GRAPH NEURAL NETWORKS

## ABSTRACT

Current graph neural network (GNN) architectures naively average or sum node embeddings into an aggregated graph representation—potentially losing structural or semantic information. We here introduce OT-GNN, a model that computes graph embeddings using parametric prototypes that highlight key facets of different graph aspects. Towards this goal, we are (to our knowledge) the first to successfully combine optimal transport (OT) with parametric graph models. Graph representations are obtained from Wasserstein distances between the set of GNN node embeddings and "prototype" point clouds as free parameters. We theoretically prove that, unlike traditional sum aggregation, our function class on point clouds satisfies a fundamental universal approximation theorem. Empirically, we address an inherent collapse optimization issue by proposing a noise contrastive regularizer to steer the model towards truly exploiting the optimal transport geometry. Finally, we consistently report better generalization performance on several molecular property prediction tasks, while exhibiting smoother graph representations.

## 1 INTRODUCTION

Recently, there has been considerable interest in developing learning algorithms for structured data such as graphs. For example, molecular property prediction has many applications in chemistry and drug discovery (Yang et al., 2019; Vamathevan et al., 2019). Historically, graphs were decomposed into features such as molecular fingerprints, or turned into non-parametric graph kernels (Vishwanathan et al., 2010; Shervashidze et al., 2011). More recently, learned representations via graph neural networks (GNNs) have achieved state-of-the-art on graph prediction tasks (Duvenaud et al., 2015; Defferrard et al., 2016; Kipf & Welling, 2017; Yang et al., 2019).

Despite these successes, graph neural networks are often underutilized in whole graph prediction tasks such as molecule property prediction. Specifically, while GNNs produce node embeddings for each atom in the molecule, these are typically aggregated via simple operations such as a sum or average, turning the molecule into a single vector prior to classification or regression. As a result, some of the information naturally extracted by node embeddings may be lost.

Departing from this simple aggregation step, Togninalli et al. (2019) recently proposed a kernel function over graphs by directly comparing non-parametric node embeddings as point clouds through optimal transport (Wasserstein distance). Their *non-parametric* model yields better empirical performance over popular graph kernels, but this idea hasn't been extended to the more challenging parametric case where optimization difficulties have to be reconciled with the combinatorial aspects of optimal transport solvers.

Motivated by these observations and drawing inspiration from prior work on prototype learning (appendix F), we introduce a new class of GNNs where the key representational step consists of comparing each input graph to a set of abstract prototypes (fig. 1). These prototypes play the role of basis functions; they are stored as point clouds as if they were encoded from actual real graphs. Each input graph is first encoded into a set of node embeddings using any existing GNN architecture. We then compare this resulting embedding point cloud to the prototype embedding sets. Formally, the distance between two point clouds is measured by their optimal transport Wasserstein distances. The prototypes as abstract basis functions can be understood as keys that highlight property values associated with different graph structural features. In contrast to previous kernel methods, the prototypes are learned together with the GNN parameters in an end-to-end manner.

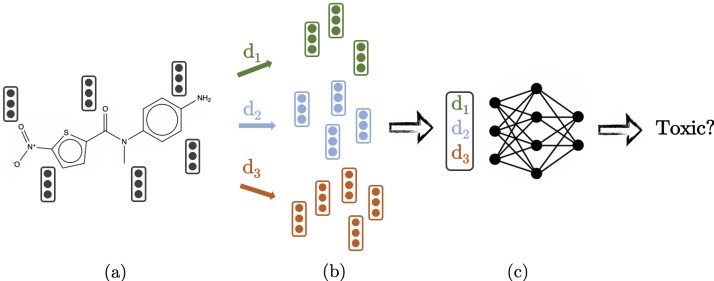

Figure 1: Our OT-GNN prototype model computes graph embeddings from Wasserstein distances between (a) the set of GNN node embeddings and (b) prototype embedding sets. These distances are then used as the molecular representation (c) for supervised tasks, e.g. property prediction. We assume that a few prototypes, e.g. some functional groups, highlight key facets or structural features of graphs relevant to a particular downstream task at hand. We express graphs by relating them to these abstract prototypes represented as free point cloud parameters.

**Our notion of prototypes** is inspired from the vast prior work on prototype learning which we highlight in appendix F. In our case, prototypes are not required to be the mean of a cluster of data, but instead they are entities living in the data embedding space that capture helpful information for the task under consideration. The closest analogy are the centers of radial basis function networks (Chen et al., 1991; Poggio & Girosi, 1990), but we also inspire from learning vector quantization approaches (Kohonen, 1995) and prototypical networks (Snell et al., 2017).

Our model improves upon traditional aggregation by explicitly tapping into the full set of node embeddings without collapsing them first to a single vector. We theoretically prove that, unlike standard GNN aggregation, our model defines a class of set functions that is a universal approximator.

Introducing prototype points clouds as free parameters trained using combinatorial optimal transport solvers creates a challenging optimization problem. Indeed, as the models are trained end-to-end, the primary signal is initially available only in aggregate form. If trained as is, the prototypes often collapse to single points, reducing the Wasserstein distance between point clouds to Euclidean comparisons of their means. To counter this effect, we introduce a contrastive regularizer which effectively prevents the model from collapsing (Section 3.2). We demonstrate its merits empirically.

**Our contributions.** First, we introduce an efficiently trainable class of graph neural networks enhanced with optimal transport (OT) primitives for computing graph representations based on relations with abstract prototypes. Second, we are the first to successfully train parametric graph models together with combinatorial OT distances, despite optimization difficulties. A key element is our noise contrastive regularizer that prevents the model from collapsing back to standard summation, thus fully exploiting the OT geometry. Third, we provide a theoretical justification of the increased representational power compared to the standard GNN aggregation method. Finally, our model shows consistent empirical improvements over previous state-of-the-art on molecular datasets, yielding also smoother graph embedding spaces.

## 2 PRELIMINARIES

### 2.1 DIRECTED MESSAGE PASSING NEURAL NETWORKS (D-MPNN)

We briefly remind here of the simplified D-MPNN (Dai et al., 2016) architecture which was adapted for state-of-the-art molecular property prediction by Yang et al. (2019). This model takes as input a directed graph $G = (V, E)$, with node and edge features denoted by $\mathbf{x}_v$ and $\mathbf{e}_{vw}$ respectively, for $v$, $w$ in the vertex set $V$ and $v \to w$ in the edge set $E$. The parameters of D-MPNN are the matrices $\{\mathbf{W}_i, \mathbf{W}_m, \mathbf{W}_o\}$. It keeps track of *messages* $\mathbf{m}_{vw}^t$ and *hidden states* $\mathbf{h}_{vw}^t$ for each step $t$, defined as follows. An initial hidden state is set to $\mathbf{h}_{vw}^0 := ReLU(\mathbf{W}_i \text{cat}(\mathbf{x}_v, \mathbf{e}_{vw}))$ where "cat" denotes concatenation. Then, the updates are:

$$\mathbf{m}_{vw}^{t+1} = \sum_{k \in N(v) \backslash \{w\}} \mathbf{h}_{kv}^t, \quad \mathbf{h}_{vw}^{t+1} = ReLU(\mathbf{h}_{vw}^0 + \mathbf{W}_m \mathbf{m}_{vw}^{t+1}), \tag{1}$$

where $N(v) = \{k \in V | (k, v) \in E\}$ denotes $v$'s incoming neighbors. After $T$ steps of message passing, node embeddings are obtained by summing edge embeddings:

$$\mathbf{m}_v = \sum_{w \in N(v)} \mathbf{h}_{vw}^T, \quad \mathbf{h}_v = ReLU(\mathbf{W}_o \text{cat}(\mathbf{x}_v, \mathbf{m}_v)). \quad (2)$$

A final graph embedding is then obtained as $\mathbf{h} = \sum_{v \in V} \mathbf{h}_v$, which is usually fed to a multilayer perceptron (MLP) for classification or regression.

## 2.2 Optimal Transport Geometry

Optimal Transport (Peyré et al., 2019) is a mathematical framework that defines distances or similarities between objects such as probability distributions, either discrete or continuous, as the cost of an optimal transport plan from one to the other.

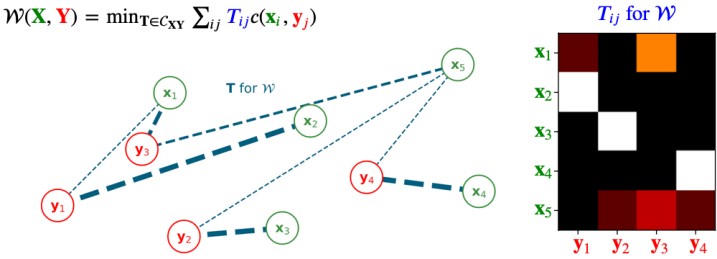

Figure 2: We illustrate, for a given 2D point cloud, the optimal transport plan obtained from minimizing the Wasserstein costs; $c(\cdot, \cdot)$ denotes the Euclidean distance. A higher dotted-line thickness illustrates a greater mass transport.

**Wasserstein distance for point clouds.** Let a *point cloud* $\mathbf{X} = \{\mathbf{x}_i\}_{i=1}^n$ *of size* $n$ be a set of $n$ points $\mathbf{x}_i \in \mathbb{R}^d$. Given point clouds $\mathbf{X}, \mathbf{Y}$ of respective sizes $n, m$, a ***transport plan*** (or ***coupling***) is a matrix $\mathbf{T}$ of size $n \times m$ with entries in $[0, 1]$, satisfying the two following *marginal constraints*: $\mathbf{T}\mathbf{1}_m = \frac{1}{n}\mathbf{1}_n$ and $\mathbf{T}^T\mathbf{1}_n = \frac{1}{m}\mathbf{1}_m$. Intuitively, the marginal constraints mean that $\mathbf{T}$ preserves the mass from $\mathbf{X}$ to $\mathbf{Y}$. We denote the set of such couplings as $\mathcal{C}_{\mathbf{XY}}$.

Given a cost function $c$ on $\mathbb{R}^d$, its associated ***Wasserstein discrepancy*** is defined as

$$\mathcal{W}(\mathbf{X}, \mathbf{Y}) = \min_{\mathbf{T} \in \mathcal{C}_{\mathbf{XY}}} \sum_{ij} T_{ij} c(\mathbf{x}_i, \mathbf{y}_j). \quad (3)$$

We further describe the shape of optimal transports for point clouds of same sizes in Appendix B.3.

## 3 Model & Practice

### 3.1 Architecture Enhancement

**Reformulating standard architectures.** As mentioned at the end of Section 2.1, the final graph embedding $\mathbf{h} = \sum_{v \in V} \mathbf{h}_v$ obtained by aggregating node embeddings is usually fed to a MLP performing a matrix-multiplication whose i-th component is $(\mathbf{R}\mathbf{h})_i = \langle \mathbf{r}_i, \mathbf{h} \rangle$, where $\mathbf{r}_i$ is the i-th row of matrix $\mathbf{R}$. Replacing $\langle \cdot, \cdot \rangle$ by a distance/kernel $k(\cdot, \cdot)$ allows the processing of more general graph representations than just vectors in $\mathbb{R}^d$, such as point clouds or adjacency tensors.

**From a single point to a point cloud.** We propose to replace the aggregated graph embedding $\mathbf{h} = \sum_{v \in V} \mathbf{h}_v$ by the point cloud (of unaggregated node embeddings) $\mathbf{H} = \{\mathbf{h}_v\}_{v \in V}$, and the inner-products $\langle \mathbf{h}, \mathbf{r}_i \rangle$ by the below written ***Wasserstein discrepancy***:

$$\mathcal{W}(\mathbf{H}, \mathbf{Q}_i) := \min_{\mathbf{T} \in \mathcal{C}_{\mathbf{HQ}_i}} \sum_{vj} T_{vj} c(\mathbf{h}_v, \mathbf{q}_i^j), \quad (4)$$

where $\mathbf{Q}_i = \{\mathbf{q}_i^j\}_{j \in \{1,\dots,N\}}, \forall i \in \{1, \dots, M\}$ represent $M$ prototype point clouds each being represented as a set of $N$ embeddings as free trainable parameters, and the cost is chosen as $c = \|\cdot - \cdot\|_2^2$ or $c = -\langle \cdot, \cdot \rangle$. Note that both options yield identical optimal transport plans.

**Greater representational power.** We formulate mathematically in Section 4 that this kernel has a strictly greater representational power than the kernel corresponding to standard inner-product on top of a sum aggregation, to distinguish between different point clouds.

**Final architecture.** Finally, the vector of all Wasserstein distances in eq. (4) becomes the input to a final MLP with a single scalar as output. This can then be used as the prediction for various downstream tasks. This model is depicted in fig. 1 and motivated theoretically in section 4.1.

## 3.2 Contrastive Regularization

What would happen to $\mathcal{W}(\mathbf{H}, \mathbf{Q}_i)$ if all points $\mathbf{q}_i^j$ belonging to point cloud $\mathbf{Q}_i$ would collapse to the same point $\mathbf{q}_i$? All transport plans would yield the same cost, giving for $c = -\langle \cdot, \cdot \rangle$:

$$\mathcal{W}(\mathbf{H}, \mathbf{Q}_i) = -\sum_{vj} T_{vj} \langle \mathbf{h}_v, \mathbf{q}_i^j \rangle = -\langle \mathbf{h}, \mathbf{q}_i / |V| \rangle. \tag{5}$$

In this scenario, our proposition would simply over-parametrize the standard Euclidean model.

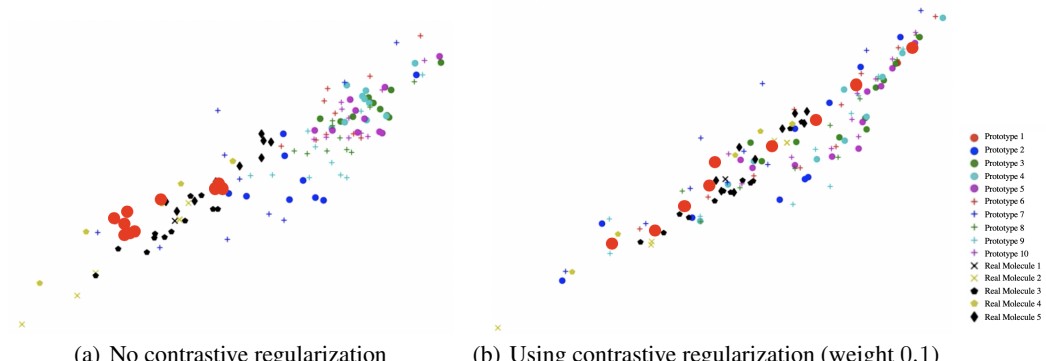

(a) No contrastive regularization      (b) Using contrastive regularization (weight 0.1)

Figure 3: 2D embeddings of prototypes and of real molecule samples with (right) and without (left) contrastive regularization for runs using the same random seed. Points in the prototypes tend to cluster and collapse more when no regularization is used, suggesting that the optimal transport plan no longer remains uniquely discriminative. Prototype 1 (red) is enlarged for clarity: without regularization it is clumped together (left), but with regularization it is distributed across the space (right).

**A first obstacle and its cause.** Our first empirical trials with OT-enhanced GNNs showed that a model trained with only the Wasserstein component would sometimes perform similarly to the Euclidean baseline in both train and validation settings, in spite of its greater representational power.

Further investigation revealed that the Wasserstein model would naturally displace the points in each of its prototype point clouds in such a way that the optimal transport plan $\mathbf{T}$ obtained by maximizing $\sum_{vj} T_{vj} \langle \mathbf{h}_v, \mathbf{q}_i^j \rangle$ was not *discriminative*, *i.e.* many other transports would yield a similar Wasserstein cost. Indeed, as shown in Eq. (5), if each point cloud collapses to its mean, then the Wasserstein geometry collaspses to Euclidean geometry. In this scenario, any transport plan yields the same Wasserstein cost. Further explanations are provided in Appendix A.1 and Figure 3..

**Contrastive regularization.** To address this difficulty, we consider adding a regularizer which encourages the model to displace its prototype point clouds such that the optimal transport plans would be discriminative against chosen *contrastive transport plans*. Namely, consider a point cloud $\mathbf{Y}$ of node embeddings and let $\mathbf{T}^i$ be an optimal transport plan obtained in the computation of $\mathcal{W}(\mathbf{Y}, \mathbf{Q}_i)$. For each $\mathbf{T}^i$, we then build a set $Neg(\mathbf{T}^i) \subset \mathcal{C}_{\mathbf{YQ}_i}$ of *noisy/contrastive* transports. If we denote by $\mathcal{W}_{\mathbf{T}}(\mathbf{X}, \mathbf{Y}) := \sum_{kl} T_{kl} c(\mathbf{x}_k, \mathbf{y}_l)$ the Wasserstein cost obtained for the particular transport $\mathbf{T}$, then our contrastive regularization consists in maximizing the term:

$$\sum_i \log \left( \frac{e^{-\mathcal{W}_{\mathbf{T}^i}(\mathbf{Y}, \mathbf{Q}_i)}}{e^{-\mathcal{W}_{\mathbf{T}^i}(\mathbf{Y}, \mathbf{Q}_i)} + \sum_{\mathbf{T} \in Neg(\mathbf{T}^i)} e^{-\mathcal{W}_{\mathbf{T}}(\mathbf{Y}, \mathbf{Q}_i)}} \right), \tag{6}$$

which can be interpreted as the log-likelihood that the correct transport $\mathbf{T}_i$ be (as it should) a better minimizer of $\mathcal{W}_{\mathbf{T}}(\mathbf{Y}, \mathbf{Q}_i)$ than its negative samples. This can be considered as an approximation of $\log(\Pr(\mathbf{T}_i \mid \mathbf{Y}, \mathbf{Q}_i))$, where the partition function is approximated by our selection of negative examples, as done e.g. by Nickel & Kiela (2017). Its effect is shown in Figure 3.

**Remarks.** The selection of negative examples should reflect the trade-off: *(i)* to not be too large, for computational efficiency while *(ii)* containing sufficiently meaningful and challenging contrastive samples. Details about choice of contrastive samples are given in Appendix A.2. Note that replacing the set $Neg(\mathbf{T}^i)$ with a singleton $\{\mathbf{T}\}$ for a contrastive random variable $\mathbf{T}$ lets us rewrite Eq. (6) as[1] $\sum_i \log \sigma(\mathcal{W}_{\mathbf{T}} - \mathcal{W}_{\mathbf{T}^i})$, reminiscent of noise contrastive estimation (Gutmann & Hyvärinen, 2010).

### 3.3 Optimization & Complexity

Backpropagating gradients through optimal transport (OT) has been the subject of recent research investigations: Genevay et al. (2017) explain how to unroll and differentiate through the Sinkhorn procedure solving OT, which was extended by Schmitz et al. (2018) to Wasserstein barycenters. However, more recently, Xu (2019) proposed to simply invoke the envelop theorem (Afriat, 1971) to support the idea of keeping the optimal transport plan fixed during the back-propagation of gradients through Wasserstein distances. *For the sake of simplicity and training stability, we resort to the latter procedure: keeping $\mathbf{T}$ fixed during back-propagation.* We discuss complexity in appendix C.

## 4 Theoretical Analysis

In this section we show that the standard architecture lacks a fundamental property of *universal approximation* of functions defined on point clouds, and that our proposed architecture recovers this property. We will denote by $\mathcal{X}_d^n$ the set of point clouds $\mathbf{X} = \{\mathbf{x}_i\}_{i=1}^n$ of size $n$ in $\mathbb{R}^d$.

### 4.1 Universality

As seen in Section 3.1, we have replaced the sum aggregation $-$ followed by the Euclidean inner-product $-$ by Wasserstein discrepancies. How does this affect the function class and representations?

A common framework used to analyze the geometry inherited from similarities and discrepancies is that of kernel theory. A kernel $k$ on a set $\mathcal{X}$ is a symmetric function $k : \mathcal{X} \times \mathcal{X} \to \mathbb{R}$, which can either measure similarities or discrepancies. An important property of a given kernel on a space $\mathcal{X}$ is whether simple functions defined on top of this kernel can approximate any continuous function on the same space. This is called *universality*: a crucial property to regress unknown target functions.

**Universal kernels.** A kernel $k$ defined on $\mathcal{X}_d^n$ is said to be ***universal*** if the following holds: for any compact subset $\mathcal{X} \subset \mathcal{X}_d^n$, the set of functions in the form[2] $\sum_{j=1}^m \alpha_j \sigma(k(\cdot, \theta_j) + \beta_j)$ is dense in the set $\mathcal{C}(\mathcal{X})$ of continuous functions from $\mathcal{X}$ to $\mathbb{R}$, w.r.t the sup norm $\| \cdot \|_{\infty, \mathcal{X}}$, $\sigma$ denoting the sigmoid. Although the notion of universality does not indicate how easy it is in practice to learn the correct function, it at least guarantees the absence of a fundamental bottleneck of the model using this kernel.

In the following we compare the aggregating kernel $\mathfrak{agg}(\mathbf{X}, \mathbf{Y}) := \langle \sum_i \mathbf{x}_i, \sum_j \mathbf{y}_j \rangle$ (used by popular GNN models) with the Wasserstein kernel, where

$$\mathcal{W}_{\mathrm{L2}}(\mathbf{X}, \mathbf{Y}) := \min_{\mathbf{T} \in \mathcal{C}_{\mathbf{XY}}} \sum_{ij} T_{ij} \|\mathbf{x}_i - \mathbf{y}_j\|_2^2, \ \mathcal{W}_{\mathrm{dot}}(\mathbf{X}, \mathbf{Y}) := \max_{\mathbf{T} \in \mathcal{C}_{\mathbf{XY}}} \sum_{ij} T_{ij} \langle \mathbf{x}_i, \mathbf{y}_j \rangle. \quad (7)$$

**Theorem 1.** *We have that:*

1. *The aggregation kernel $\mathfrak{agg}$ **is not universal**.*

2. *The Wasserstein kernel $\mathcal{W}_{\mathrm{L2}}$ **is universal**.*

*Proof:* See appendix B.1. Universality of the $\mathcal{W}_{\mathrm{L2}}$ kernel comes from the fact that its square-root defines a metric, and from the axiom of separation of distances: *if $d(x, y) = 0$ then $x = y$.*

---

[1] where $\sigma(\cdot)$ is the sigmoid function.
[2] For $m \in \mathbb{N}^*$, $\alpha_j \beta_j \in \mathbb{R}$ and $\theta_j \in \mathcal{X}_d^n$.

**Implications.** Theorem 1 states that our proposed OT-GNN model is strictly more powerful than the state of the art GNN models that use summation or averaging of node embeddings. Nevertheless, this implies we can only distinguish graphs that have distinct multi-sets of node embeddings, e.g. all Weisfeiler-Lehman distinguishable graphs in the case of graph convolutional networks.

In practice, the shape of the aforementioned functions that posses universal approximation capabilities gives an indication of how one should leverage the vector of Wasserstein distances to prototypes to perform graph classification – e.g. using a multilayer perceptron (MLP) on top.

## 4.2 DEFINITENESS

For the sake of simplified mathematical analysis, similarity kernels are often required to be *positive definite* (p.d.), which corresponds to discrepancy kernels being *conditionally negative definite* (c.n.d.). Although such a property has the benefit of yielding the mathematical framework of Reproducing Kernel Hilbert Spaces, it essentially implies *linearity*, *i.e.* the possibility to embed the geometry defined by that kernel in a linear vector space.

We now discuss that, interestingly, the Wasserstein kernel we used does not satisfy this property, and hence constitutes an interesting instance of a universal, non p.d. kernel. Let us remind these notions.

**Kernel definiteness.** A kernel $k$ is ***positive definite (p.d.)*** on $\mathcal{X}$ if for $n \in \mathbb{N}^*$, $x_1, ..., x_n \in \mathcal{X}$ and $c_1, ..., c_n \in \mathbb{R}$, we have $\sum_{ij} c_i c_j k(x_i, x_j) \geq 0$. It is ***conditionally negative definite (c.n.d.)*** on $\mathcal{X}$ if for $n \in \mathbb{N}^*$, $x_1, ..., x_n \in \mathcal{X}$ and $c_1, ..., c_n \in \mathbb{R}$ such that $\sum_i c_i = 0$, we have $\sum_{ij} c_i c_j k(x_i, x_j) \leq 0$.

These two notions relate to each other via the below result Boughorbel et al. (2005):

**Proposition 1.** *Let $k$ be a symmetric kernel on $\mathcal{X}$, let $x_0 \in \mathcal{X}$ and define the kernel:*

$$\tilde{k}(x, y) := -\frac{1}{2}[k(x, y) - k(x, x_0) - k(y, x_0) + k(x_0, x_0)].$$ (8)

*Then $\tilde{k}$ **is p.d. if and only if** $k$ is c.n.d. Example: $k = \| \cdot - \cdot \|_2^2$ and $x_0 = \boldsymbol{0}$ yield $\tilde{k} = \langle \cdot, \cdot \rangle$.*

One can easily show that $\mathfrak{agg}$ also defines a p.d. kernel, and that $\mathfrak{agg}(\cdot, \cdot) \leq n^2 \mathcal{W}(\cdot, \cdot)$. However, the Wasserstein kernel is not p.d., as stated in different variants before (e.g. Vert (2008)) and as reminded by the below theorem. We here give a novel proof in Appendix B.2.

**Theorem 2.** *We have that:*

1. *The (similarity) Wasserstein kernel $\mathcal{W}_{\text{dot}}$ **is not positive definite**;*

2. *The (discrepancy) Wasserstein kernel $\mathcal{W}_{\text{L2}}$ **is not conditionally negative definite**.*

## 5 EXPERIMENTS

### 5.1 EXPERIMENTAL SETUP

We experiment on 4 benchmark molecular property prediction datasets (Yang et al., 2019) including both regression (ESOL, Lipophilicity) and classification (BACE, BBBP) tasks. These datasets cover different complex chemical properties (e.g. ESOL - water solubility, LIPO - octanol/water distribution coefficient, BACE - inhibition of human $\beta$-secretase, BBBP - blood-brain barrier penetration).

We provide results for our implementation of 4 different widely used graph-based models.

**Fingerprint + MLP** applies a MLP over the input features which are hashed graph structures (called a molecular fingerprint). **GIN** is the Graph Isomorphism Network from (Xu et al., 2019), which is a variant of a GNN. The original GIN does not account for edge features, so we adapt their algorithm to our setting. Next, **GAT** is the Graph Attention Network from (Veličković et al., 2017), which uses multi-head attention layers to propagate information. The original GAT model does not account for edge features, so we adapt their algorithm to our setting. More details about our implementation of the GIN and GAT models can be found in the appendix D.2. Finally, **Chemprop - D-MPNN** (Yang et al., 2019) is a graph network that exhibits state-of-the-art performance for molecular representation learning across multiple classification and regression datasets. Empirically

we find that this baseline is indeed the best performing, and so is used as to obtain node embeddings in all our prototype models. Its architecture is described in section 2.1.

Different variants of our OT-GNN prototype model are described below:

**ProtoW-L2/Dot** is the model that treats point clouds as point sets, and computes the Wasserstein distances to each point cloud (using either L2 distance or (minus) dot product cost functions) as the molecular embedding. **ProtoS-L2** is a special case of **ProtoW-L2**, in which the point clouds have a *single* point and instead of using Wasserstein distances, we just compute simple Euclidean distances between the aggregated graph embedding and point clouds. Here, we omit using dot product distances, as that model is mathematically equivalent to the GNN model.

We use the the POT library (Flamary & Courty, 2017) to compute Wasserstein distances using the network simplex algorithm (Earth Movers distance), which we find empirically to be faster than the Sinkhorn algorithm for our datasets. We define the cost matrix by taking the pairwise L2 or negative dot product distances. As mentioned in Section 3.3, we fix the transport plan, and only backpropagate through the cost matrix for computational efficiency. Additionally, to account for the variable size of each input graph, we multiply the OT distance between two point clouds by their respective sizes. To avoid the problem of point clouds collapsing, we employ the contrastive regularizer defined in Section 3.2. More details about experimental setup are presented in Appendix D.1.

| | Models | ESOL (RMSE) # grphs = 1128 | Lipo (RMSE) # grphs= 4199 | BACE (AUC) # grphs= 1512 | BBBP (AUC) # grphs= 2039 |
|---|---|---|---|---|---|
| Baselines | Fingerprint + MLP | .922 ± .017 | .885 ± .017 | .870 ± .007 | .911 ± .005 |
| | GIN | .665 ± .026 | .658 ± .019 | .861 ± .013 | .900 ± .014 |
| | GAT | .654 ± .028 | .808 ± .047 | .860 ± .011 | .888 ± .015 |
| | D-MPNN | .635 ± .027 | .646 ± .041 | .865 ± .013 | .915 ± .010 |
| Ours | ProtoS-L2 | .611 ± .034 | **.580 ± .016** | .865 ± .010 | .918 ± .009 |
| | ProtoW-Dot *(no reg.)* | .608 ± .029 | .637 ± .018 | .867 ± .014 | .919 ± .009 |
| | ProtoW-Dot | **.594 ± .031** | .629 ± .015 | .871 ± .014 | .919 ± .009 |
| | ProtoW-L2 *(no reg.)* | .616 ± .028 | .615 ± .025 | .870 ± .012 | **.920 ± .010** |
| | ProtoW-L2 | .605 ± .029 | .604 ± .014 | **.873 ± .015** | **.920 ± .010** |

Table 1: Results on the property prediction datasets. **Best** model is in bold, second best is underlined. Lower RMSE is better, while higher AUC is better. Wasserstein models are by default trained with contrastive regularization as described in section 3.2 and outperform those without.

## 5.2 EXPERIMENTAL RESULTS

### 5.2.1 REGRESSION AND CLASSIFICATION

Results are shown in Table 1. Our prototype models outperform state-of-the art GNN/D-MPNN baselines on all 4 property prediction tasks. Moreover, the prototype models using Wasserstein distance (**ProtoW-L2/Dot**) achieve better performance on 3 out of 4 of the datasets compared to the prototype model using only Euclidean distances (**ProtoS-L2**). This indicates that Wasserstein distance confers greater discriminative power compared to traditional aggregation methods.

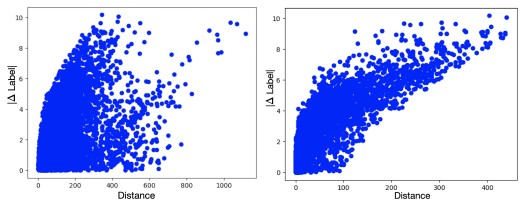

(a) D-MPNN model    (b) ProtoW-L2 model

Figure 4: Comparison of the correlation between graph embedding distances (X axis) and label distances (Y axis) on the ESOL dataset.

Table 2: The Spearman and Pearson correlation coefficients on the ESOL dataset for the GNN and ProtoW-L2 model w.r.t. the pairwise difference in embedding vectors and labels.

| | Spearman $\rho$ | Pearson $r$ |
|---|---|---|
| D-MPNN | .424 ± .029 | .393 ± .049 |
| ProtoS-L2 | .561 ± .087 | .414 ± .141 |
| ProtoW-Dot | .592 ± .150 | .559 ± .216 |
| ProtoW-L2 | **.815 ± .026** | **.828 ± .020** |

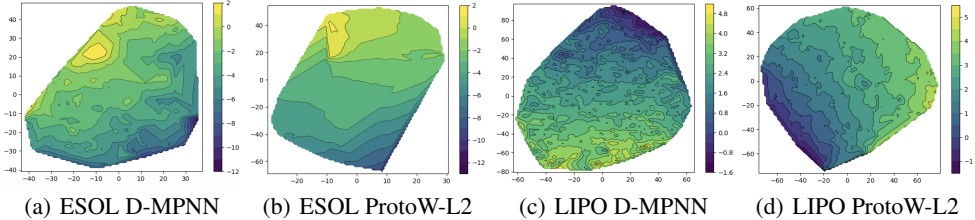

(a) ESOL D-MPNN     (b) ESOL ProtoW-L2     (c) LIPO D-MPNN     (d) LIPO ProtoW-L2

Figure 5: 2D heatmaps of T-SNE Maaten & Hinton (2008) projections of molecular embeddings (before the last linear layer) w.r.t. their associated predicted labels. Heat colors are interpolations based only on the test molecules from each dataset. Comparing (a) vs (b) and (c) vs (d), we can observe a smoother space of our model compared to the D-MPNN baseline (see also main text).

### 5.2.2 NOISE CONTRASTIVE REGULARIZER

Without any constraints, the Wasserstein prototype model will often collapse the set of points in a point cloud into a single point. As mentioned in Section 3.2, we use a contrastive regularizer to force the model to meaningfully distribute point clouds in the embedding space. We show 2D embeddings in Fig. 3, illustrating that without contrastive regularization, prototype point clouds are often displaced close to their mean, while regularization forces them to nicely scatter. Quantitative results in Table 1 also highlight the benefit of this regularization.

### 5.2.3 LEARNED EMBEDDING SPACE: QUALITATIVE AND QUANTITATIVE RESULTS

We further examine the learned embedding space of the best baseline (i.e. D-MPNN) and our best Wasserstein model. We claim that our models learn smoother latent representations. We compute the pairwise difference in embedding vectors and the labels for each test data point on the ESOL dataset. Then, we compute two measures of rank correlation, Spearman correlation coefficient ($\rho$) and Pearson correlation coefficient ($r$). This is reminiscent of evaluation tasks for the correlation of word embedding similarity with human labels (Luong et al., 2013).

Our ProtoW-L2 achieves better $\rho$ and $r$ scores compared to the D-MPNN model (Table 2), that indicating our Wasserstein model constructs more meaningful embeddings with respect to the label distribution. Indeed, Figure 4 plots the pairwise scores for the D-MPNN model (left) and the ProtoW-L2 model (right). Our ProtoW-L2 model, trained to optimize distances in the embedding space, produces more meaningful representations with respect to the label of interest.

Moreover, as qualitatively shown in Figure 5, our model provides more robust molecular embeddings compared to the baseline, in the following sense: we observe that a small perturbation of a molecular embedding corresponds to a small change in predicted property value – a desirable phenomenon that holds rarely for the baseline D-MPNN model. Our Proto-W-L2 model yields smoother heatmaps.

| | Average Train Time per Epoch (sec) | | | | Average Total Train Time (min) | | | |
|---|---|---|---|---|---|---|---|---|
| | D-MPNN | ProtoS | ProtoW | ProtoW + NC reg | D-MPNN | ProtoS | ProtoW | ProtoW + NC reg |
| ESOL | 5.5 | 5.4 | 13.5 | 31.7 | 4.3 | 5.3 | 15.4 | 30.6 |
| BACE | 32.0 | 31.4 | 25.9 | 39.7 | 12.6 | 12.4 | 25.1 | 49.4 |
| BBBP | 41.8 | 42.4 | 51.2 | 96.4 | 12.6 | 13.2 | 31.5 | 62.2 |
| LIPO | 33.3 | 35.5 | 27 | 69.4 | 28.6 | 31.8 | 74.7 | 146.9 |

Table 3: Training times for each model and dataset.

| Model | Fingerprint + MLP | GIN | GAT | D-MPNN | ProtoS | ProtoW |
|---|---|---|---|---|---|---|
| # Parameters | 401k | 626k | 671k | 100k | 65k | 66k |

Table 4: Number of parameters per model. Corresponding hyperparameters are in appendix D.1.

### 5.2.4 Additional Experimental Details

**Model Sizes**    Using MPNN hidden dimension as 200, and the final output MLP hidden dimension as 100, the number of parameters for the models are given by table 4. The fingerprint used dimension was 2048, explaining why the MLP has a large number of parameters. The D-MPNN model is much smaller than GIN and GAT models because it shares parameters between layers, unlike the others. Our prototype models are even smaller than the D-MPNN model because we do not require the large MLP at the end, instead we compute distances to a few small prototypes (small number of overall parameters used for these point clouds). The dimensions of the prototype embeddings are also smaller compared to the node embedding dimensions of the D-MPNN and other baselines. We did not see significant improvements in quality by increasing any of the hyperparameter values.

Remarkably, our model outperforms all the baselines using between 10 to 1.5 times less parameters.

**Runtimes.**    We report the average total training time (number of epochs might vary depending on the early stopping criteria), as well as average training epoch time for the D-MPNN and our prototype models in table 3. We note that our method is between 1 to 7.1 times slower than the D-MPNN baseline which mostly happens due to the frequent calls to the Earth Mover Distance OT solver.

## 6 Related Work

Graph Neural Networks were introduced by Gori et al. (2005) and Scarselli et al. (2008) as a form of recurrent neural networks. Graph convolutional networks (GCN) appeared later on in various forms. Duvenaud et al. (2015); Atwood & Towsley (2016) proposed a propagation rule inspired from convolution and diffusion, but these methods do not scale to graphs with either large degree distribution or node cardinality. Niepert et al. (2016) defined a GCN as a 1D convolution on a chosen node ordering. Kearnes et al. (2016) also used graph convolutions to generate high quality molecular fingerprints. Efficient spectral methods were proposed by Bruna et al. (2013); Defferrard et al. (2016). Kipf & Welling (2017) simplified their propagation rule, motivated from spectral graph theory (Hammond et al., 2011). Different such architectures were later unified into the message passing neural networks framework by Gilmer et al. (2017), and applied to molecular property prediction. A directed variant of message passing from Dai et al. (2016) was later used to improve state-of-the-art in molecular property prediction on a wide variety of datasets by (Yang et al., 2019). Other applications include recommender systems (Ying et al., 2018a). Inspired by DeepSets Zaheer et al. (2017), Xu et al. (2019) suggest a simplified architecture called GIN, which theoretically can discriminate between any different local neighborhoods. Other recent approaches modify the sum-aggregation of node embeddings in the GCN architecture with the aim to preserve more information Kondor et al. (2018); Pei et al. (2020). In this category there is also the recently growing class of hierarchical graph pooling methods which typically either use deterministic and non-differentiable node clustering (Defferrard et al., 2016; Jin et al., 2018), or differentiable pooling (Ying et al., 2018b; Noutahi et al., 2019; Gao & Ji, 2019). However, these methods are still strugling with small labelled graphs such as molecules where global and local node interconnections cannot be easily cast as a hierarchical interaction.

Other recent geometry-inspired GNNs include adaptations to non-Euclidean spaces (Liu et al., 2019; Chami et al., 2019; Bachmann et al., 2019).

We additionally discuss related work on prototype learning in appendix F.

## 7 Conclusion

We propose **OT-GNN**: one of the first parametric graph models that leverages optimal transport to learn graph representations. It learns abstract prototypes as free parametric point clouds that highlight different aspects of the graph. Empirically, we outperform popular baselines in different molecular property prediction tasks, while the learned representations also exhibit stronger correlation with the target labels. Finally, universal approximation theoretical results enhance the merits of our model.

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

## A    FURTHER DETAILS ON CONTRASTIVE REGULARIZATION

### A.1    MOTIVATION

One may speculate that it was locally easier for the model to extract valuable information if it would behave like the Euclidean component, preventing it from exploring other roads of the optimization landscape. To better understand this situation, consider the scenario in which a subset of points in a prototype point cloud "collapses", i.e. become close to each other (see Figure 3), thus sharing similar distances to all the node embeddings of real input graphs. The submatrix of the optimal transport matrix corresponding to these collapsed points can be equally replaced by any other submatrix with the same marginals (*i.e.* same two vectors obtained by summing rows or columns), meaning that the optimal transport matrix is not discriminative. In general, we want to avoid any two rows or columns in the Wasserstein cost matrix being proportional. An additional problem of point collapsing is that it is a non-escaping situation when using gradient-based learning methods. The reason is that gradients of these collapsed points would become and remain identical, thus nothing will encourage them to "separate" in the future.

### A.2    ON THE CHOICE OF CONTRASTIVE SAMPLES

Our experiments were conducted with ten negative samples for each correct transport plan. Five of them were obtained by initializing a matrix with uniform *i.i.d* entries from $[0, 10)$ and performing around five Sinkhorn iterations (Cuturi, 2013) in order to make the matrix satisfy the marginal constraints. The other five were obtained by randomly permuting the columns of the correct transport plan. The latter choice has the desirable effect of penalizing the points of a prototype point cloud $\mathbf{Q}_i$ to collapse onto the same point. Indeed, the rows of $\mathbf{T}^i \in \mathcal{C}_{\mathbf{HQ}_i}$ index points in $\mathbf{H}$, while its columns index points in $\mathbf{Q}_i$.

## B    THEORETICAL RESULTS

### B.1    PROOF OF THEOREM 1

**1.**    Let us first justify why $\mathfrak{agg}$ is not universal. Consider a function $f \in \mathcal{C}(\mathcal{X})$ such that there exists $\mathbf{X}, \mathbf{Y} \in \mathcal{X}$ satisfying both $f(\mathbf{X}) \neq f(\mathbf{Y})$ and $\sum_k \mathbf{x}_k = \sum_l \mathbf{y}_l$. Clearly, any function of the form $\sum_i \alpha_i \sigma(\mathfrak{agg}(\mathbf{W}_i, \cdot) + \theta_i)$ would take equal values on $\mathbf{X}$ and $\mathbf{Y}$ and hence would not approximate $f$ arbitrarily well.

**2.**    To justify that $\mathcal{W}$ is universal, we take inspiration from the proof of universality of neural networks Cybenko (1989).

**Notation.**    Denote by $M(\mathcal{X})$ the space of finite, signed regular Borel measures on $\mathcal{X}$.

**Definition.**    We say that $\sigma$ is discriminatory w.r.t a kernel $k$ if for a measure $\mu \in M(\mathcal{X})$,

$$\int_{\mathcal{X}} \sigma(k(\mathbf{Y}, \mathbf{X}) + \theta) d\mu(\mathbf{X}) = 0$$

for all $\mathbf{Y} \in \mathcal{X}_d^n$ and $\theta \in \mathbb{R}$ implies that $\mu \equiv 0$.

We start by reminding a lemma coming from the original paper on the universality of neural networks by Cybenko Cybenko (1989).

**Lemma.**    If $\sigma$ is discriminatory w.r.t. $k$ then $k$ is universal.

*Proof:* Let $S$ be the subset of functions of the form $\sum_{i=1}^m \alpha_i \sigma(k(\cdot, \mathbf{Q}_i) + \theta_i)$ for any $\theta_i \in \mathbb{R}$, $\mathbf{Q}_i \in \mathcal{X}_d^n$ and $m \in \mathbb{N}^*$ and denote by $\bar{S}$ the closure[3] of $S$ in $\mathcal{C}(\mathcal{X})$. Assume by contradiction that

---

[3]W.r.t the topology defined by the sup norm $\|f\|_{\infty, \mathcal{X}} := \sup_{X \in \mathcal{X}} |f(X)|$.

$\bar{S} \neq \mathcal{C}(\mathcal{X})$. By the Hahn-Banach theorem, there exists a bounded linear functional $L$ on $\mathcal{C}(\mathcal{X})$ such that for all $h \in \bar{S}$, $L(h) = 0$ and such that there exists $h' \in \mathcal{C}(\mathcal{X})$ s.t. $L(h') \neq 0$. By the Riesz representation theorem, this bounded linear functional is of the form:

$$L(h) = \int_{\mathbf{X} \in \mathcal{X}} h(\mathbf{X}) d\mu(\mathbf{X}),$$

for all $h \in \mathcal{C}(\mathcal{X})$, for some $\mu \in M(\mathcal{X})$. Since $\sigma(k(\mathbf{Q}, \cdot) + \theta)$ is in $\bar{S}$, we have

$$\int_{\mathcal{X}} \sigma(k(\mathbf{Q}, \mathbf{X}) + \theta) d\mu(\mathbf{X}) = 0$$

for all $\mathbf{Q} \in \mathcal{X}_d^n$ and $\theta \in \mathbb{R}$. Since $\sigma$ is discriminatory w.r.t. $k$, this implies that $\mu = 0$ and hence $L \equiv 0$, which is a contradiction with $L(h') \neq 0$. Hence $\bar{S} = \mathcal{C}(\mathcal{X})$, *i.e.* $S$ is dense in $\mathcal{C}(\mathcal{X})$ and $k$ is universal.

$\square$

Now let us look at the part of the proof that is new.

**Lemma.** $\sigma$ is discriminatory w.r.t. $\mathcal{W}_{\mathrm{L2}}$.

*Proof:* Note that for any $\mathbf{X}, \mathbf{Y}, \theta, \varphi$, when $\lambda \to +\infty$ we have that $\sigma(\lambda(\mathcal{W}_{\mathrm{L2}}(\mathbf{X}, \mathbf{Y}) + \theta) + \varphi)$ goes to 1 if $\mathcal{W}_{\mathrm{L2}}(\mathbf{X}, \mathbf{Y}) + \theta > 0$, to 0 if $\mathcal{W}_{\mathrm{L2}}(\mathbf{X}, \mathbf{Y}) + \theta < 0$ and to $\sigma(\varphi)$ if $\mathcal{W}_{\mathrm{L2}}(\mathbf{X}, \mathbf{Y}) + \theta = 0$.

Denote by $\Pi_{\mathbf{Y}, \theta} := \{\mathbf{X} \in \mathcal{X} \mid \mathcal{W}_{\mathrm{L2}}(\mathbf{X}, \mathbf{Y}) - \theta = 0\}$ and $B_{\mathbf{Y}, \theta} := \{\mathbf{X} \in \mathcal{X} \mid \sqrt{\mathcal{W}_{\mathrm{L2}}(\mathbf{X}, \mathbf{Y})} < \theta\}$ for $\theta \geq 0$ and $\emptyset$ for $\theta < 0$. By the Lebesgue Bounded Convergence Theorem we have:

$$0 = \int_{\mathbf{X} \in \mathcal{X}} \lim_{\lambda \to +\infty} \sigma(\lambda(\mathcal{W}_{\mathrm{L2}}(\mathbf{X}, \mathbf{Y}) - \theta) + \varphi) d\mu(\mathbf{X})$$
$$= \sigma(\varphi)\mu(\Pi_{\mathbf{Y}, \theta}) + \mu(\mathcal{X} \setminus B_{\mathbf{Y}, \sqrt{\theta}}).$$

Since this is true for any $\varphi$, it implies that $\mu(\Pi_{\mathbf{Y}, \theta}) = \mu(\mathcal{X} \setminus B_{\mathbf{Y}, \sqrt{\theta}}) = 0$. From $\mu(\mathcal{X}) = 0$ (because $B_{\mathbf{Y}, \sqrt{\theta}} = \emptyset$ for $\theta < 0$), we also have $\mu(B_{\mathbf{Y}, \sqrt{\theta}}) = 0$. Hence $\mu$ is zero on all balls defined by the metric $\sqrt{\mathcal{W}_{\mathrm{L2}}}$.

From the Hahn decomposition theorem, there exist disjoint Borel sets $P, N$ such that $\mathcal{X} = P \cup N$ and $\mu = \mu^+ - \mu^-$ where $\mu^+(A) := \mu(A \cap P)$, $\mu^-(A) := \mu(A \cap N)$ for any Borel set $A$ with $\mu^+, \mu^-$ being positive measures. Since $\mu^+$ and $\mu^-$ coincide on all balls on a finite dimensional metric space, they coincide everywhere Hoffmann-Jørgensen (1976) and hence $\mu \equiv 0$.

$\square$

Combining the previous lemmas with $k = \mathcal{W}_{\mathrm{L2}}$ concludes the proof.

$\square$

## B.2 PROOF OF THEOREM 2

**1.** We build a counter example. We consider 4 point clouds of size $n = 2$ and dimension $d = 2$. First, define $\mathbf{u}_i = (\lfloor i/2 \rfloor, i\%2)$ for $i \in \{0, ..., 3\}$. Then take $\mathbf{X}_1 = \{\mathbf{u}_0, \mathbf{u}_1\}$, $\mathbf{X}_2 = \{\mathbf{u}_0, \mathbf{u}_2\}$, $\mathbf{X}_3 = \{\mathbf{u}_0, \mathbf{u}_3\}$ and $\mathbf{X}_4 = \{\mathbf{u}_1, \mathbf{u}_2\}$. On the one hand, if $\mathcal{W}(\mathbf{X}_i, \mathbf{X}_j) = 0$, then all vectors in the two point clouds are orthogonal, which can only happen for $\{i, j\} = \{1, 2\}$. On the other hand, if $\mathcal{W}(\mathbf{X}_i, \mathbf{X}_j) = 1$, then either $i = j = 3$ or $i = j = 4$. This yields the following Gram matrix

$$(\mathcal{W}(\mathbf{X}_i, \mathbf{X}_j))_{0 \leq i,j \leq 3} = \begin{pmatrix} 1 & 0 & 1 & 1 \\ 0 & 1 & 1 & 1 \\ 1 & 1 & 2 & 1 \\ 1 & 1 & 1 & 2 \end{pmatrix} \tag{9}$$

whose determinant is $-1/16$, which implies that this matrix has a negative eigenvalue.

**2.** This comes from proposition 1. Choosing $k = \mathcal{W}_{\text{L2}}$ and $x_0 = \mathbf{0}$ to be the trivial point cloud made of $n$ times the zero vector yields $\tilde{k} = \mathcal{W}_{\text{dot}}$. Since $\tilde{k}$ is not positive definite from the previous point of the theorem, $k$ is not conditionally negative definite from proposition 1.

$\square$

### B.3 SHAPE OF THE OPTIMAL TRANSPORT PLAN FOR POINT CLOUDS OF SAME SIZE

The below result describes the shape of optimal transport plans for point clouds of same size. For the sake of curiosity, we also illustrate in Figure 2 the optimal transport for point clouds of different sizes. We note that non-square transports seem to remain relatively sparse as well. This is in line with our empirical observations.

**Proposition 2.** *For $\mathbf{X}, \mathbf{Y} \in \mathcal{X}_{n,d}$ there exists a rescaled permutation matrix $\frac{1}{n}(\delta_{i\sigma(j)})_{1 \leq i,j \leq n}$ which is an optimal transport plan, i.e.*

$$\mathcal{W}_{\text{L2}}(\mathbf{X}, \mathbf{Y}) = \frac{1}{n} \sum_{j=1}^{n} \|\mathbf{x}_{\sigma(j)} - \mathbf{y}_j\|_2^2, \quad \mathcal{W}_{\text{dot}}(\mathbf{X}, \mathbf{Y}) = \frac{1}{n} \sum_{j=1}^{n} \langle \mathbf{x}_{\sigma(j)}, \mathbf{y}_j \rangle. \tag{10}$$

*Proof.* It is well known from Birkhoff's theorem that every squared doubly-stochastic matrix is a convex combination of permutation matrices. Since the Wasserstein cost for a given transport $\mathbf{T}$ is a linear function, it is also a convex/concave function, and hence it is maximized/minimized over the convex compact set of couplings at one of its extremal points, namely one of the rescaled permutations, yielding the desired result. $\square$

## C COMPLEXITY

### C.1 WASSERSTEIN

Computing the Wasserstein optimal transport plan between two point clouds consists in the minimization of a linear function under linear constraints. It can either be performed exactly by using network simplex methods or interior point methods as done by (Pele & Werman, 2009) in time $\tilde{\mathcal{O}}(n^3)$, or approximately up to $\varepsilon$ via the Sinkhorn algorithm (Cuturi, 2013) in time $\tilde{\mathcal{O}}(n^2/\varepsilon^3)$. More recently, (Dvurechensky et al., 2018) proposed an algorithm solving OT up to $\varepsilon$ with time complexity $\tilde{\mathcal{O}}(\min\{n^{9/4}/\varepsilon, n^2/\varepsilon^2\})$ via a primal-dual method inspired from accelerated gradient descent.

In our experiments, we used the Python Optimal Transport (POT) library (Flamary & Courty, 2017). We noticed empirically that the Earth Mover Distance (EMD) solver yielded faster and more accurate solutions than Sinkhorn for our datasets, because the graphs and point clouds were small enough ($< 30$ elements). However, Sinkhorn may take the lead for larger graphs.

Significant speed up could potentially be obtained by rewritting the POT library for it to solve OT in batches over GPUs. In our experiments, we ran all jobs on CPUs.

# D    Further Experimental Details

## D.1    Setup of Experiments

Each dataset is split randomly 5 times into 80%:10%:10% train, validation and test sets. For each of the 5 splits, we run each model 5 times to reduce the variance in particular data splits (resulting in each model being run 25 times). We search hyperparameters for each split of the data, and then take the average performance over all the splits. The hyperparameters are separately searched for each data split, so that the model performance is based on a completely unseen test set, and that there is no data leakage across data splits. The models are trained for 150 epochs with early stopping if the validation error has not improved in 50 epochs and a batch size of 16. We train the models using the Adam optimizer with a learning rate of 5e-4. For the prototype models, we use different learning rates for the GNN and the point clouds (5e-4 and 5e-3 respectively), because empirically we find that the gradients are much smaller for the point clouds. The molecular datasets used for experiments here are small in size (varying from 1-4k data points), so this is a fair method of comparison, and is indeed what is done in other works on molecular property prediction Yang et al. (2019).

| Parameter Name | Search Values | Description |
|---|---|---|
| n_epochs | $\{150\}$ | Number of epochs trained |
| batch_size | $\{16\}$ | Size of each batch |
| lr | $\{5e\text{-}4\}$ | Overall learning rate for model |
| lr_pc | $\{5e\text{-}3\}$ | Learning rate for the prototype embeddings |
| n_layers | $\{5\}$ | Number of layers in the GNN |
| n_hidden | $\{50, 200\}$ | Size of hidden dimension in GNN |
| n_ffn_hidden | $\{1e2, 1e3, 1e4\}$ | Size of the output feed forward layer |
| dropout_gnn | $\{0.\}$ | Dropout probability for GNN |
| dropout_fnn | $\{0., 0.1, 0.2\}$ | Dropout probability for feed forward layer |
| n_pc (M) | $\{10, 20\}$ | Number of the prototypes (point clouds) |
| pc_size (N) | $\{10\}$ | Number of points in each prototype (point cloud) |
| pc_hidden (d) | $\{5, 10\}$ | Size of hidden dimension of each point in each prototype |
| nc_coef | $\{0., 0.01, 0.1, 1\}$ | Coefficient for noise contrastive regularization |

Table 5: The parameters for our models (the prototype models all use the same GNN base model), and the values that we used for hyperparameter search. When there is only a single value in the search list, it means we did not search over this value, and used the specified value for all models.

## D.2    Baseline models

Both the GIN (Xu et al., 2019) and GAT (Veličković et al., 2017) models were originally used for graphs without edge features. Therefore, we adapt both these algorithms to our use-case, in which edge features are a critical aspect of the prediction task. Here, we expand on the exact architectures that we use for both models.

First we introduce common notation that we will use for both models. Each example is defined by a set of vertices and edges $(V, E)$. Let $v_i \in V$ denote the $i$th node in the graph, and let $e_{ij} \in E$ denote the edge between nodes $(i, j)$. Let $h_{v_i}^{(k)}$ be the feature representation of node $v_i$ at layer $k$. Let $h_{e_{ij}}$ be the input features for the edge between nodes $(i, j)$, and is static because we do updates only on nodes. Let $N(v_i)$ denote the set of neighbors for node $i$, not including node $i$; let $\hat{N}(v_i)$ denote the set of neighbors for node $i$ as well as the node itself.

**GIN**

The update rule for GIN is then defined as:

$$h_{v_i}^k = \text{MLP}^{(k)}\Big((1 + \epsilon^{(k)}) + \sum_{v_j \in N(v_i)} [h_u^{(k-1)} + W_b^{(k)} h_{e_{ij}}]\Big) \qquad (11)$$

As with the original model, the final embedding $h_G$ is defined as the concatenation of the summed node embeddings for each layer.

$$h_G = \text{CONCAT}\Big[\sum_{v_i} h_{v_i}^{(k)} | k = 0, 1, 2...K\Big] \tag{12}$$

**GAT**

For our implementation of the GAT model, we compute the attention scores for each pairwise node $\alpha_{ij}^{(k)}$ as follows.

$$\alpha_{ij}^{(k)} = \frac{\exp\Big(\text{LeakyReLU}(a^{(k)}\Big[W_1^{(k)}h_{v_i}^{(k-1)}||W_2^{(k)}[h_{v_j}^{(k-1)} + W_b^{(k)}h_{e_{ij}}]\Big])\Big)}{\sum_{v_j \in \hat{N}(v_i)} \exp\Big(\text{LeakyReLU}(a^{(k)}\Big[W_1^{(k)}h_{v_i}^{(k-1)}||W_2^{(k)}[h_{v_j}^{(k-1)} + W_b^{(k)}h_{e_{ij}}]\Big])\Big)} \tag{13}$$

$\{W_1^{(k)}, W_2^{(k)}, W_b^{(k)}\}$ are layer specific feature transforms, while $a^{(k)}$ is a vector that computes the final attention score for each pair of nodes. Note that here we do attention across all of a node's neighbors as well as the node itself.

The updated node embeddings are as follows:

$$h_{v_i}^k = \sum_{v_j \in \hat{N}(v_i)} \alpha_{i,j}^{(k)} h_{v_j}^{(k-1)} \tag{14}$$

The final graph embedding is just a simple sum aggregation of the node representations on the last layer ($h_G = \sum_{v_i} h_{v_i}^K$). As with (Veličković et al., 2017), we also extend this formulation to utilize multiple attention heads.

## E    WHAT TYPES OF MOLECULES DO PROTOTYPES CAPTURE ?

To better understand if the learned prototypes offer interpretability, we examined the ProtoW-Dot model trained with NC regularization (weight 0.1). For each of the 10 learned prototypes, we computed the set of molecules in the test set that are closer in terms of the corresponding Wasserstein distance to this prototype than to any other prototype. While we noticed that one prototype is closest to the majority of molecules, there are other prototypes that are more interpretable as shown in fig. 6.

## F    RELATED WORK ON PROTOTYPE LEARNING

Learning prototypes to solve machine learning tasks has been extensively studied. Generalized learning vector quantization (GLVQ) methods (Kohonen, 1995; Sato & Yamada, 1995) perform classification by assigning to each data point the class of the closest neighbor prototype measured using some distance function, typically Euclidean. Each class has a set of prototypes that are optimized together such that the closest wrong prototype is moved away, while the correct prototype is brought closer. Several extensions of GLVQ (Hammer & Villmann, 2002; Schneider et al., 2009; Bunte et al., 2012) introduce feature weights and parameterized input transformations to leverage more flexible and adaptive metric spaces. Nevertheless, such models are limited to classification tasks and might suffer from extreme gradient sparsity.

On the other hand, more similar to our work are the radial basis function (RBF) networks (Chen et al., 1991) that perform classification or regression based on RBF kernel similarities to all prototypes. One such similarity vector is computed for each data point and used together with a shared linear output layer to obtain the final predictions. Prototypes are typically set in an unsupervised fashion, e.g. via k-means clustering, or using the Orthogonal Least Square Learning algorithm, unlike being learned using backpropagation as in our case.

Combining the non-parametric mathematical power of kernel methods with the flexibility of deep learning models have resulted in even more expressive and scalable similarity functions that have

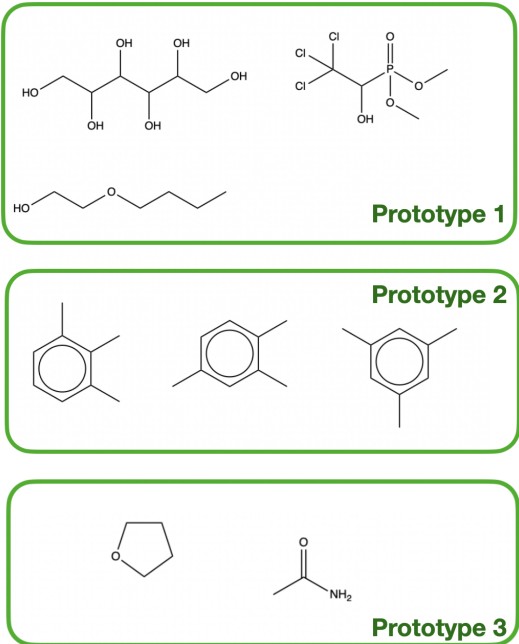

Figure 6: The closest molecules to some particular prototypes in terms of the corresponding Wasser-stein distance. One can observe that some prototypes are closer to insoluble molecules containing rings (Prototype 2), while others prefer more soluble molecules (Prototype 1).

been conveniently trained with backpropagation by maximizing the likelihood of a Gaussian process (Wilson et al., 2016). Recently, learning parametric data embeddings and prototypes was also investigated for few-shot and zero-shot classification scenarios (Snell et al., 2017).

Finally, using distances to prototypes as opposed to p.d. kernels, while not as common, was analyzed in the past (Duin & Pękalska, 2012; Snell et al., 2017).

In contrast with the above line of work, our research focuses on learning parametric prototypes for graphs trained jointly with graph embedding functions for both graph classification and regression problems. Prototypes are modeled as sets (point clouds) of embeddings, while graphs are represented by sets of unaggregated node embeddings obtained using graph neural network models. Disimilarities between prototypes and graph embeddings are then quantified via set distances computed using optimal transport. Additional challenges arise due to the combinatorial nature of the Wasserstein distances between sets, hence our discussion on introducing the noise contrastive regularizer.

