# OpenReview forum: "Optimal Transport Graph Neural Networks"
_ICLR.cc/2021/Conference — Reject_

### Official Review · AnonReviewer4 · 2020-10-13
**A strong combination of optimal transport and graph neural nets, with some room for improvement with respect to prototypes**

**Rating:** 7
**Confidence:** 5

**Review:**

## Summary

The paper introduces a novel approach to aggregate information of graph neural network node embeddings in order to support graph-level machine learning, like graph classification or graph regression. The aggregation is performed by comparing the node embeddings of a graph to learned, prototypical node embeddings via the Wasserstein distance. The vector of distances to the prototypes then serves as a concise representation that can be fed into a subsequent standard multi-layer perceptron for classification or regression. This approach is proven to be strictly more powerful than just adding up the node embeddings, which is the most common state-of-the-art. Finally, the paper evaluates the proposed aggregation function on two graph classification and two graph regression data sets, showing superior performance in all cases.

## Strengths

The paper has several strengths worth highlighting:

* The idea of using prototypical node embeddings as a method for aggregation is easy to understand and potentially interpretable. For example, one could try to construct graphs that best correspond to each prototype and thereby gather understanding what typical graphs look like based on which the model makes its decision.
* The idea also corresponds well to past research on kernels as well as distances, thus nicely integrating the strengths of graph neural nets and kernel/distance theory. The use of the Wasserstein distance to compare point clouds is particularly compelling and combines two hot topics in machine learning today, namely graph neural nets and the Wasserstein distance, which should be interesting to a broad section of the ICLR community.
* The theoretical results, especially the universal approximation result, are compelling and further make clear that the present work is strictly more powerful than the state of the art (i.e. just summing up all node embeddings)
* The experimental evaluation is performed against multiple reasonable baselines, including ablation variants of the proposed model, on several data sets, covering both regression and classification problems. Multiple repeats are performed, giving an impression of variance. Additionally, for the regression data sets, the paper shows both correlation as well as dimensionality reduction results indicating that the proposed representation is more smooth in relation to the targets compared to existing models.

## Weaknesses

That being said, I also see some issues which could be addressed to make the paper even stronger.

* Most importantly, from my perspective, the relation to existing work warrants much more discussion in two directions:
    * Currently, the discussion of prototype learning is reduced to the work of Snell (2017), whereas a long research tradition of learning prototypes for classification and regression is currently not covered. In particular, learning vector quantization approaches (refer e.g. to [Sato & Yamada; 1995][GLVQ]) have been used to learn prototypes that represent classes well and can then be used for a one-nearest neighbor classification. Even more similar to the present paper, radial basis function networks (e.g. [Chen, Cowan, and Grant, 1991][RBF]) have been used to solve regression tasks in the space of RBF kernel values to prototypes. It would be, I believe, helpful to contrast the present prototype learning approach to these past works. Additionally, these past works may give inspiration how to learn prototypes well and how to solve the issues of prototype collapse.
    * Currently, the paper treats distances as a kind of non-definite kernel. This can be done, but is unusual. An understanding of kernel and distance as opposing (but related) concepts is more common and, I believe, more instructive. However, the use of distances as features can still be justified, e.g. using the work of [Pekalska & Duin (2012)][DistanceTheory]. Similarly, it would be helpful to relate the present work to attempts that build bridges between kernels and neural nets, such as [Wilson, Hu, Salakhutdinov, and Xing (2016)][DeepKernels]
* I see several issues with how prototypes are integrated into the model.
    * It is not clear to me why the distances to the prototypes are plugged into another MLP instead of using them directly, at least for classification. In particular, prior work of Snell (2017) as well as [learning vector quantization][GLVQ] suggests to associate each prototype with a label and learn the prototypes such that a one-nearest-neighbor classification solves the classification problem. For regression, the issue is a bit more difficult, but here, as well, I would expect a single additional layer, similar to [RBF networks][RBF] and not several additional layers. Ideally, it would be beneficial to provide comparisons to these alternatives as additional ablation results in the appendix.
    * Using prototypes introduces two new hyperparameters, namely the number of prototypes and the size of points within each prototype. The paper currently gives no guidance on how to choose these hyperparameters.
    * Prototypes have the advantage of being a sparse representation of the data which lends itself to interpretation. However, currently no such interpretation is attempted. This seems to me like a missed opportunity. Showing an example graph for each prototype (i.e. a graph that is close to one prototype but far from every other prototype) could give additional insights into how the model represents the data.
    * The regularizer suggested in the paper does not seem sufficiently justified to me, yet. In particular, there are exponentially many 'bad' transport plans which I would not like to choose. How can I guarantee that my set N(T_i) is sufficiently rich to 'protect' me against all bad transport plans? Conversely, if the aim is just to prevent collapsing of the prototypes, would there not be simpler ways to prevent such a collapse, such as regularizing with the negative log of pairwise distances between points inside a prototype?
* Leaving the optimal transport plan T constant during backpropagation seems reasonable to me but the justification appears still rather hand-wavy (this applies to the cited paper of Xu (2019) as well). After inspecting the original paper of Afriat (1971), I am not ale to see how the results described there are applicable to transport plans (I don't even see an 'envelop theorem' in there). The connection may be there, but it is not obvious to me. Going back to a higher-level intuition, I think the key issue here is that even a very slight change in the points of a prototype set could dramatically change the optimal transport plan (that is just how transport plans behave, is my understanding; whenever another assignment becomes slightly better, suddenly the assignment changes in a discontinuous fashion). This speaks against leaving it constant. In the worst case, it could occur that the current transport plan suggests to move a point into one direction, but the transport plan at the next point suggests to move it back, yielding an endless cycle. One could, however, approximate the transport plan smoothly, e.g. as a mixture of several transport plans between which we interpolate continuously, and then one could make the argument that small gradient steps do not change the plan much, hence we can leave it constant. But again: Such an argument is not trivial and is currently missing. Furthermore, it would be good to clarify that the transport plan is only treated as constant for backpropagation, but that it is updated after each gradient step (that is, at least, how I interpret the approach).
* It would help to qualify Theorem 1 a little. First, Theorem 1 does not imply that the Wasserstein kernel can distinguish between all graphs; only between the graphs that are distinct in the multi-set of node embeddings, i.e. all Weisfeiler-Lehman distinguishable graphs. Second, Theorem 1 does not imply that the model as proposed is a universal approximator because the proposed model only uses a finite set of prototypes to represent the data. To make it a universal approximator, one would need infinitely many prototypes, if I am not mistaken. In practice, this is likely not an issue because few prototypes are sufficient to cover the space pretty well. It is still a limitation to how much Theorem 1 actually tells us about the proposed model.
* The experiments currently do not report empiric runtime results. These would be a valuable addition because they illustrate how much overhead prototypes introduce compared to just summing up the final layer.

There are also a few smaller issues which could be clarified:

* page 3: I wonder why the marginal distributions over points in X and points in Y are restricted to be uniform. Would the more general case not be interesting as well, where, say, prototypes receive more weight if they are more important for representing the data?
* Page 3: The vectors r_i are not introduced. Are these supposed to be the weight vectors of neurons in the MLP?
* Page 3: It is not clear to me why the negative inner product and the Euclidean distance should yield the same transport plan. I can see that for the special case of vectors with norm 1 where ||x - y||^2 = 1 - 2 * <x, y> + 1, but not in the general case.

## Judgment

Overall, I believe that this paper is strong enough to warrant acceptance. In particular, the heart of the paper, namely the proposed model architecture, the main theoretical result, and the experiments, are strong and my issues - namely related work discussions, some theoretical fine points, and ablation studies - are aspects that can be addressed during a revision without affecting the core paper too much. Furthermore, I believe that the core idea - combining prototypes, optimal transport, and graph neural nets - has great potential to make graph neural nets stronger and more interpretable, which are both crucial goals for the research community right now. As such, I am confident that the paper, as it stands, is worthy of publication and, with some additional work during revision, can become even better.

## References

* Sato, A., & Yamada, K. (1995). Generalized Learning Vector Quantization. Proceedings of NIPS 1995. [Link][GLVQ]
* de Vries, H., Memisevic, R., & Courville, A. (2016). Deep Learning Vector Quantization. Proceedings of ESANN 2016. [Link][DLVQ]
* Chen, S., Cowan, C., and Grant, P. (1991). Orthogonal least squares learning algorithm for radial basis function networks. IEEE Transactions on neural networks, 2(2). [Link][RBF]
* Duin, R., and Pekalska, E. (2012). The dissimilarity space: Bridging structural and statistical pattern recognition. Pattern Recognition Letters, 33(7). doi:[10.1016/j.patrec.2011.04.019][DistanceTheory]
* Wilson, A., Hu, Z., Salakhutdinov, R., and Xing, E. (2016). Deep Kernel Learning. Proceedings of AISTATS 2016. [Link][DeepKernels]

[GLVQ]:https://papers.nips.cc/paper/1113-generalized-learning-vector-quantization "Sato, A., & Yamada, K. (1995). Generalized Learning Vector Quantization. Proceedings of NIPS 1995."
[DLVQ]:https://www.elen.ucl.ac.be/Proceedings/esann/esannpdf/es2016-112.pdf "de Vries, H., Memisevic, R., & Courville, A. (2016). Deep Learning Vector Quantization. Proceedings of ESANN 2016."
[RBF]:https://eprints.soton.ac.uk/251135/1/00080341.pdf "Chen, S., Cowan, C., and Grant, P. (1991). Orthogonal least squares learning algorithm for radial basis function networks. IEEE Transactions on neural networks, 2(2)."
[DistanceTheory]:https://doi.org/10.1016/j.patrec.2011.04.019 "Duin, R., and Pekalska, E. (2012). The dissimilarity space: Bridging structural and statistical pattern recognition. Pattern Recognition Letters, 33(7)."
[DeepKernels]:http://proceedings.mlr.press/v51/wilson16.html "Wilson, A., Hu, Z., Salakhutdinov, R., and Xing, E. (2016). Deep Kernel Learning. Proceedings of AISTATS 2016"

## Clarifications

While I am quite confident in my review, it would help if the authors could verify that I understood some key points in the paper correctly:

* Each prototype here is a collection of points, and the number of prototypes as well as the number of points is a fixed hyperparameter, whereas the location of each point is learned, correct?
* The pipeline for classification/regression is to first embed the nodes of a graph via a graph neural net, yielding a point cloud with one point per node; second, to compare this point cloud to all prototypes using the Wasserstein distance/inner product; third to collect the resulting distances/inner products into a vector d/k with as many entries as there are prototypes and to feed this vector into a standard MLP to make the final classification/regression decision, correct?
* The optimal transport plan T is treated as constant for backprop, but is recomputed after each gradient step, correct?
* Are the limitations to Theorem 1 described above correct or did I misunderstand the Theorem?

## Typos

Beyond the points mentioned above, there are also a few typos and very minor things that do not affect my judgment but could easily be fixed to improve the paper.

* Abstract: The abbreviation 'OT' is not introduced
* page 1: 'As a result, some of the information naturally extracted by node embeddings may be lost' <-- This is correct, but could be underlined more with a citation
* page 2: 'that is universal approximator.' -> that is a universal approximator
* page 2: The abbreviation 'D-MPNN' is not introduced
* Figure 3 is currently rather busy and not very easy to interpret. Would it, perhaps, suffice to only show prototype 1 and one reference molecule?
* As far as I know, ICLR permits an appendix. As such, it would be good to move the appendix into the main paper.

---

> ### Author Response · Authors · 2020-11-16
> **Thank you for your detailed suggestions!**
>
> We thank the reviewer for the very detailed and insightful comments and suggestions. We reply below:
>
> - **Related work on prototypes:**  Thank you for all the suggestions. We have added a related work section about prototype learning in Appendix F.
>
> - **Prototypes:**
>   - **MLP on top of vector of distances:** Theorem 1 provides a hint on how the vector of distances to prototypes could be used to perform graph classification tasks. We theoretically show in section 4.1 that the function should have the shape $\sum_{j=1}^m \alpha_j \sigma(k(\cdot,\theta_j) + \beta_j)$ which explains our choice of applying an MLP on top of the Wasserstein distance vector in order to perform classification. We updated section 4.1 and discussed practical implications of our theory. We also tried using a linear layer instead of a MLP, but did not see significant gains.
>
>   - **New hyperparameters:** As mentioned in appendix D.1, we treat these hyperparameters like usual, doing grid search over a few choices shown in Table 5.
>
>   - **Interpretability:** We added some examples of molecules closest to some prototypes in appendix E and figure 6. We indeed observe that some prototypes prefer molecules containing rings, while others prefer more soluble and cycle-free graphs.
>
>   - **Choosing negative transport plans in the NC regularizer:**  As discussed in the main text and appendix A, we desire to choose negative transport plans that both are very close to the true transport plan, as well as cover the space of all possible transport plans. Our choice of negative examples is also very efficient in practice.
>
>   - **Alternative regularizer:** Before deriving the NC regularization, we did try variants of your proposed regularizer such as increasing the variance of pairwise distances inside a prototype. However, in all these cases the model was still clumping together some of the points, while moving others undesirably far. Our proposed NC regularizer is motivated by the need of the optimal transport plan to be sufficiently discriminative among all other valid transport plans, which implies that conglomerates of prototype points are discouraged.
>
> - **Transport plan being fixed during backpropagation:** We first confirm that the transport plan is only treated as constant for backpropagation, but recomputed by calling the EMD routine for each forward pass. Next, we note that the loss, as a function of the transport plan, has zero gradients almost everywhere, i.e. the set of points where it has non-zero sub-gradients is a null measure set. For small enough updates, this amounts to neglecting backpropagation w.r.t. the transport plan with probability 1. Regarding the smooth approximation of the transport plan as a mixture of several transport plans, this could be done but it would be non-trivial to decide what are the basic transport plans to be interpolated (since these matrices are typically very sparse). An alternative is to relax the doubly stochastic constraints and use a parametric model similar as in [2]. We leave this direction for future work.
>
> - **Theorem 1:** We fully agree that Theorem 1 implies we can only distinguish graphs that have distinct multisets of node embeddings, i.e. all Weisfeiler-Lehman distinguishable graphs. We updated our section 4.1 to discuss implications of our theoretical result. See also our reply to Reviewer 2. Regarding the second point, we agree with the reviewer and note that, in practice, 10 or 20 prototypes suffice for the datasets we considered.
>
> - **Running times** As suggested by the reviewers, we added all training times (total and per epoch) for all models and the best baseline in Table 3 and discussed it in section 5.2.4 . Our models are indeed sometimes slower than the baselines which is expected because of the frequent calls to the Earth Mover Distance algorithm for computing the optimal transport plan. Please also see replies to other reviewers.
>
> - **Non-uniform marginals:** We agree that using unbalanced optimal transport solvers would definitely be interesting as an extension of our work. There are however a number of new challenges such as adapting existing unbalanced OT solvers (that, in our experience, suffer from various numerical errors), as well as preventing new types of model collapse cases. We leave extensive exploration of this direction as future work.
>
> - **Notation of r_i:** fixed.
>
> [2] Object-Centric Learning with Slot Attention, F. Locatello et al, NeurIPS 2020

---

> > ### Author Response · Authors · 2020-11-16
> > **Last comments**
> >
> > **Why W-L2 and W-dot give the same transport plan**:
> > We have $W_{L2}(X,Y) := \min_{T\in \mathcal{C}(X,Y)}\sum_{ij} T_{ij} || x_i-y_j ||_2^2$ and
> >
> > $W_{dot}(X,Y) :=  \max_{T\in \mathcal{C}(X,Y)} \sum_{ij} T_{ij}\langle x_i,y_j \rangle$. We can rewrite the former as follows: $W_{L2}(X,Y) = \frac{1}{m}\sum_i ||x_i||^2 + \frac{1}{n}\sum_j ||y_j||^2 - \max_{T}  \sum_{ij} T_{ij}\langle x_i,y_j \rangle $, where $n$ and $m$ are the sizes of the X and Y point clouds. The last relation happens because the matrix $T$ sums on each row to $\frac{1}{n}$ and on each column to $\frac{1}{m}$. More details on this computation can be read in e.g. [1].
> >
> >
> > **Clarification questions:** Yes, all your statements and questions are exactly correct. We changed the paper to make these points clearer.
> >
> > **Typos:** Fixed.
> >
> > [1] Gromov-Wasserstein Averaging of Kernel and Distance Matrices, G. Peyre et al, ICML 2016

---

> > > ### Comment · AnonReviewer4 · 2020-11-19
> > > **Commendable response**
> > >
> > > I would like to express my gratitude for the exhaustive response provided by the authors. Indeed, the authors have clarified all my points and I have nothing left to add. I am now highly confident that this paper is worthy of acceptance.

---

### Official Review · AnonReviewer1 · 2020-10-27
**Review for OT-GNN**

**Rating:** 5
**Confidence:** 4

**Review:**

This paper introduces OT-GNN, a combination of optimal transport and graph neural network, for graph-level tasks. Instead of the conventional graph embedding+MLP strategy, the authors propose to replace the readout layer in MPNN-like structures with Wasserstein discrepancy. In addition, the noise contrastive regularizer is added to the model so that the optimal transport plan is discriminative, and the new model could outperform its Euclidean counterpart. While the construction is sophisticated, the core idea behind the algorithm is not hard to follow. However, I do have some concerns regarding the presentation and experiments, which, at this stage, prevent me from recommending the paper confidently to the broader community.


Below I would like to list my main concerns or confusions regarding the paper.

The construction of prototypes is not very clear to me. I would expect the authors to provide more discussion before the experiment section regarding its structure, requirements, or formulation details.
The authors, if my understanding is correct, named the same subject to different terms as ‘prototypes’ and ‘free parameters’. This indeed creates unnecessary hurdles of understanding the paper, and I would recommend the names could be identified, or at least clearly and properly defined.
The design of Figure 3 is confusing. What is the difference between Prototype 1-5 and Prototype 6-10? As they are designed with different shapes, I would expect them to represent different kinds of prototypes. However, unless I missed something, I didn’t find a clear statement of it. Also, I’m not sure what is the purpose of showing five real molecules? Are we supposed to compare them with the corresponding prototypes?
What is the partition function you mentioned in the 3rd line after Eq.6? Do you use different terminology from in Nickel & Kiela (2017)?
The authors only compared their method with naive node embedding+sum aggregation models but ignored a major type of graph representation learning methods as graph pooling. This branch is developed rapidly in the recent 2 years, and many simple yet powerful methods have been proposed. I don’t think it’s fair to compare with the models that are specialized in node embedding tasks but neglect the real methods that are designed for graph embedding tasks.
Following the last point, I would expect more recent research to be included in the Related Work section.
Some figures could be redesigned for better presentation. For example, it’s hard to read the axis information in Figure 4.

Minor typos:

(line 2, Section 2.1) “…property prediction by Yang et al. Yang et al. (2019).”
(line 1, Section 2.2) “Optimal Transport (OT) Peyré et al. (2019) is a mathematical framework…”

Given the problems stated as above, I would suggest the paper to be carefully polished before it’s ready to publish.

---

> ### Author Response · Authors · 2020-11-16
> **Paper updated based on reviewer's suggestions**
>
> We thank the reviewer for the comments and suggestions. We reply below:
>
> - **Clarification of the construction of prototypes:** We updated the text (especially section 3.1) to clarify our model. The prototypes are M point clouds, each containing N real vectors in d dimensions that are free learnable parameters (not tight to any particular graph structure). However, they can be understood as node embedding sets of some abstract prototype graphs that capture different facets of the real graphs in the training data distribution. These parameters are learned together with the rest of the model parameters (i.e. of the D-MPNN and of the final MLP).
>
> - **Confusion regarding the terms “prototype” and “free parameters”:** We updated the text to clarify the multiple confusions regarding these terms. As explained above, each prototype is a set of embeddings that are learned as free parameters not constrained to represent any particular graph structure.
>
> - **Figure 3:** There is no difference between the prototypes shown in Fig 3. They all contain 10 points/embeddings and are all trained in the same manner together with the rest of the model. We decided to show 5 different node embedding point clouds from different real molecules to reveal the fact that the NC regularizer makes them spread across the space and not clump together. This is also true for the prototype embeddings.
>
> - **Partition function in Eq. 6:** As mentioned in the text, Eq. 6 can be seen as approximating the value $\log(\mathrm{Pr}(\mathbf{T}_i\mid \mathbf{Y},\mathbf{Q}_i))$. However, the partition function involved is highly intractable (requires an expectation over doubly stochastic matrices). Inspired from Nickel & Kiela (2017) (and others) we propose to replace this partition function by a simple summation over negative examples of transport plans.
>
> - **Literature and methods on graph pooling:** We agreed and added a discussion about graph pooling methods in the Related Work section 6. That being said, in our experience, graph pooling methods did not show significant improvements over the D-MPNN baseline. This might be caused by the specific types of graphs that we focus on (e.g. small molecules with up to 100 nodes). We leave a more comprehensive comparison with these methods for future work.
>
> - **Figure 4 font size:** Is now fixed.
>
> - **Other modifications we did:** Following the reviewers suggestions, we added training runtimes, comparison of the models in terms of number of parameters (section 5.2.4) and discussed interpretability (appendix E). We also added clarifications to various parts of the paper such as discussing the implications of our theory.

---

### Official Review · AnonReviewer3 · 2020-10-29
**Interesting Wasserstein kernel, good empirical performance**

**Rating:** 5
**Confidence:** 3

**Review:**

This paper combines OT  with parametric graph neural network. It replace the inner product between the graph embedding and the first layer weights of MLP by the Wasserstein distance between the node embeddings and some point clouds. Then the GNN, point clouds and the downstream MLP are trained in an end-to-end way. A regularization term is adopted to enforce the point clouds are not collapsed. The authors then theoretically show that the Wasserstein kernel is universal.

Pros:
1. The empirical study is thorough and the model has good empirical performance.
2. The idea of using prototypes is interesting. That being said, I think the word "prototype" here is actually misleading. In Snell 2017, their prototypes is a representation of the data. In contrast, here Q_i is NOT the representation of H, since we are not minimizing the Wasserstein distance between them. Instead, Q_i just helps to keep some information that is useful for the task.

Areas to improve:
1. My major concern is about computation time.  In each training step, the algorithm computes batch size*(the range of indices i) of OT problems. Will the OT part introduce a large training time overhead? Could you please report the training time of each method?
2. Since the model has significantly more parameters, e.g., in Q_i, it would be better to also compare it to a baseline model with comparable number of parameters and the node embedding info. For example, is it possible to compare it with: randomly selecting k embeddings, concatenate, and feed into MLP (adjusting k to make the parameter size comparable)?
3. Some notations are not properly defined. For example, \mathfrak{agg} first appears in section 4.1, but its formal definition is in section 4.2. Another example is indices i and j in eq (4). What is the range of i and the range of j? In other words, how is the range of i, the range of j, the dimension of q_i^j corresponds to n_pc, pc_size, pc_hidden in the appendix?
4. Could you please provide some intuition that different variants of the proposed model performs the best for different dataset (table 1)? In other words, what variants is suitable for what kind of dataset?

---

> ### Author Response · Authors · 2020-11-16
> **We incorporated reviewer's suggestions**
>
> We thank the reviewer for the comments and suggestions. We reply below:
>
> 1. **Training runtimes:** As suggested by the reviewers, we added all training times (total and per epoch) for all models and the best baseline in Table 3 and discussed it in section 5.2.4 . Our models are indeed sometimes slower than the baselines which is expected because of the frequent calls to the Earth Mover Distance algorithm for computing the optimal transport plan. As noted in appendix C.1, we rely on the Python Optimal Transport (POT) library (Flamary & Courty, 2017) which is not yet optimized to solve Wasserstein distance computations in batches over GPUs. In our experiments, we ran all jobs on CPUs, but desire to move to batched GPU runs in future.
>
> 2. **OT-GNN model has significantly more parameters than the baselines:** Our proposed model and its variants do **not** have more parameters than the baseline D-MPNN or other baselines. The explanation is that few prototypes (10 or 20) are usually enough to obtain the best model quality on the validation sets, which means that our final MLP applied on the Wasserstein distances to the prototypes is much smaller than the final MLP used in D-MPNN and other baselines (which use graph embedding dimension 100 or larger).
>
>   As suggested by the reviewers, we added the number of parameters of all the models and baselines in **Table 4** and discuss in section 5.2.4.
>
>   We note that, remarkably, *our model outperforms all the baselines using between 10 to 1.5 times less parameters.*
>
> 3. **Notations:** fixed. Please see section 4, Eq. 4 and Table 5.
>
> **Other modifications:** Following the reviewers suggestions, we discussed interpretability (appendix E). We also added clarifications to various parts of the paper such as discussing the implications of our theory.

---

> ### Comment · AnonReviewer4 · 2020-11-19
> **The term 'prototype'**
>
> Because the authors did not respond to this point explicitly, I would like to mention that the word 'prototype' has been used in many different contexts in past research and it is not always required that a prototype is the mean of a cluster of data. Indeed, the rich literature on learning vector quantization approaches shows that there are meaningful ways to define prototypes in a purely discriminative context. As such, I believe that a broader definition of 'prototype' - in the sense of an entity living in the data space that captures helpful information for the task at hand - is more useful and covers the present contribution quite well.

---

> > ### Author Response · Authors · 2020-11-19
> > **We added a discussion on related work on prototype learning and clarified our notion of "prototypes"**
> >
> > Thank you for your suggestion on clarifying the term "prototypes" as used in our context. We have added a paragraph in the introduction (page 2, just below Fig 1, starts with "Our notion of prototypes"). We have also highlighted and discussed prior work on prototype learning in the related work section from Appendix F. Indeed, our notion of prototypes is not defined as a cluster center of data points, but rather as an entity that captures helpful information for the downstream task. The closest analogy is represented by the centers of radial basis function networks (Chen et al., 1991; Poggio & Girosi, 1990). We additionally note that our prototypes are designed to be utilized in both classification and regression tasks.

---

### Official Review · AnonReviewer2 · 2020-10-30
**Review for Optimal Transport Graph Neural Networks**

**Rating:** 4
**Confidence:** 4

**Review:**

**Summary**
The paper proposes OT-GNN, which incorporates optimal transport distance to message passing of GNN. The message passing is aggregated by using a Wasserstein discrepancy for a point cloud. The contrastive regularization is utilized to overcome extreme clustering of nodes of the same class. Also, in theory, the author shows that the Wasserstein kernel is universal while the "agg" kernel is not. The proposed model is tested on several molecular property prediction tasks. The OT-GNN achieves slightly better performance against existing methods.

**Pros**
1. The paper introduces a non-Euclidean metric in message-passing aggregation and develops OT-GNN based on it. The proposed OT-GNN has good performance for molecular property prediction tasks.
2. The paper provides a theoretical analysis of the universality and positiveness of the Wasserstein kernel. It shows the Wasserstein metric-induced message passing is superior to the Euclidean metric case in terms of universality. The paper also shows the $L_2$ Wasserstein kernel is not conditionally negative positive.
3. The computational complexity is analyzed for Wasserstein optimal transport in the OT-GNN.

**Cons**
1. For Wasserstein discrepancy in (4), there seem many pairs of $H$ and $Q_i$, where $Q_i$ contains a set of free parameters. Then the OT-GNN would have too many parameters to train. Will it bring about overparameterization?
2. Using the Wasserstein metric, the computational cost will increase much. Besides, the contrastive regularization will add more training time. What is the GPU wall time for OT-GNN training for the molecule regression tasks?
3. The theoretical analysis for the properties of Wasserstein kernel seems not complicated, and it does not give a good interpretation of the learning ability, like universal approximation property or generalization of OT-GNN. In particular, the discussion of positiveness of the kernel is of no significance or use.
4. The authors also need to try more public graph datasets, such as Open Graph Benchmark, https://ogb.stanford.edu/.
5. Line 4 below (6): the last sentence seems not finished.

---

> ### Author Response · Authors · 2020-11-16
> **Paper updated based on reviewer's suggestions**
>
> We thank the reviewer for the comments and suggestions. We reply below:
>
> 1. **Overparameterization and number of parameters:** Our proposed model and its variants do **not** have more parameters than the baseline D-MPNN or than other baselines. The explanation is that few prototypes (10 or 20) are usually enough to obtain the best model quality on the validation sets, which means that our final MLP applied on the Wasserstein distances to the prototypes is much smaller than the final MLP used in D-MPNN and other baselines (which use graph embedding dimension 100 or larger).
>
>   As suggested by the reviewers, we added the number of parameters of all the models and baselines in **Table 4** and discuss in section 5.2.4.
>
>   We note that, remarkably, *our model outperforms all the baselines using between 10 to 1.5 times less parameters.*
> 2. **Running times** As suggested by the reviewers, we added all training times (total and per epoch) for all models and the best baseline in Table 3 and discussed it in section 5.2.4 . Our models are indeed sometimes slower than the baselines which is expected because of the frequent calls to the Earth Mover Distance algorithm for computing the optimal transport plan. As noted in appendix C.1, we rely on the Python Optimal Transport (POT) library (Flamary & Courty, 2017) which is not yet optimized to solve Wasserstein distance computations in batches over GPUs. In our experiments, we ran all jobs on CPUs, but desire to move to batched GPU runs in future.
>
> 3. **Theory:**
>     - **Universal approximation capability of our models as implied by Theorem 1:** Our theoretical analysis does provide via theorem 1 the fundamental universal approximation property: our class of models can approximate arbitrarily well any real function on point clouds. This result is similar to the universal approximation property of deep neural networks. Importantly, this result does not hold for the aggregation/summation kernel used by most of the popular graph neural network models (e.g. D-MPNN, GAT, GIN, GCN, etc). As pointed out also by Reviewer4, this proves that our OT-GNN model is strictly more powerful than the state of the art that sums up all node embeddings. However, we also note that this implies we can only distinguish graphs that have distinct sets of node embeddings, i.e. all Weisfeiler-Lehman distinguishable graphs. We updated section 4.1 to make this clear.
>
>     - **Practical implication of Theorem 1:** Our theoretical result provides a hint on how the vector of distances to prototypes could be used to perform graph classification tasks. We theoretically show in section 4.1 that the function should have the shape $\sum_{j=1}^m \alpha_j \sigma(k(\cdot,\theta_j) + \beta_j)$ which explains our choice of applying an MLP on top of the Wasserstein distance vector in order to perform classification. We updated section 4.1 and discussed practical implications of our theory.
>
>     - **Generalization capabilities:** Inspired by recent work (e.g. [1]), we wish to explore generalization capabilities of OT-GNN models as part of our future work.
>
>     - **Discussion of kernel positiveness**: We discuss kernel positiveness to relate our approach to the rich literature on Reproducing Kernel Hilbert Spaces (RKHS). As mentioned in section 4.2, similarity kernels are often required to be positive definite (p.d.) due to the benefit of yielding the mathematical framework of RKHS, including the celebrated Representer theorem of [2]. The theoretical result shown in theorem 2 is well known (e.g. see (Vert, 2008)), but our proof is novel.
>
> 4. **OGB datasets:** The 4 datasets we used are all part of the OGB benchmark under slightly different names: ogbg-molbace, ogbg-molbbbp, ogbg-molesol, ogbg-mollipo. Please refer to [this link](https://github.com/snap-stanford/ogb/tree/master/examples/graphproppred/mol) for details. Thank you for your suggestion to try out to incorporate even more datasets which we plan to do in future.
>
> [1] Generalization and Representational Limits of Graph Neural Networks, V. Garg et al., ICML 2020
>
> [2] A Generalized Representer Theorem. Computational Learning Theory., Schölkopf et al, 2001

---

### Author Response · Authors · 2020-11-22
**Looking forward for a feedback on our improved paper**

Dear reviewers,

We believe we have addressed most of the raised issues, namely:

- **Showing training runtimes:** we have added these numbers to the paper in Table 3.

- **Concern that our models have more parameters than the baselines:** We stated that this is not a concern. In fact, *our proposed models have significantly less parameters than all the baseline models (Table 4)*, while outperforming all the baselines in all considered tasks. We added to the paper the number of parameters used by all our models and the baselines in Table 4.

- **Related work on prototypes:** We updated our related work section (appendix F) to discuss relations with the vast literature on prototype learning.

- **Related work on graph pooling methods:** We updated our related work section to discuss these methods.

- **Discussion about practical implications of our theory:** We refined the paper to discuss practical implications, e.g. choice of a 2-layer MLP on top of the Wasserstein distance vector. Besides, our Theorem 1 implies our model is a universal approximator of all real functions defined on sets of node embeddings, unlike the standard summation or aggregation kernels used in the other GNN models.

- **Typos, notations and terminology:** We have addressed all these comments in our updated version of the paper.

- **Comparison with graph pooling baselines:** D-MPNN (Yang et al., 2019) and GAT (Veličković  et al., 2017) models are state-of-the-art for molecular representation learning (which is our focus) and we do compare against these baselines. In our experience, other methods such as graph pooling do not outperform D-MPNN or GAT baselines.

We kindly ask if there is any feedback on our rebuttal or if there are still points that we need to address before the rebuttal deadline.

Thank you for your time and feedback!

---

### Decision · Program_Chairs · 2021-01-07
**Final Decision**

**Decision:**

Reject

**Comment:**

This paper proposes a very interesting approach using Wasserstein distance between graph embedded by GNN to perform prediction. The paper is well written and the experiments suggest that the method works  well in practice.

Several reviewers had some concerns about  computational complexity and model parameters that were very well answered during the discussion and with the new version of the paper but it was not enough to convince them to change their overall opinion on the paper.

Note that a lengthy discussion with the reviewers stemming from an unclear Figure 3  was done about the need for a  contrastive regularization that raised important questions that should be addressed. In short, despite the claim by the authors that the contrastive regularization is important the experiments show very little difference in performances with or without the regularization which asks the question of its usefulness. As a matter of fact looking at Figure 3 the regularization will spread the samples in the distributions, making the prototype more similar . The argument about the sample collapse is not good enough because if the sample collapse completely during the optimization the method converges toward prototype L2 which is not the case since the proposed approach is much better than L2 even without regularization. So if there is some collapse it actually serves the discrimination and leads to a better solution. Also the Wasserstsein can reverse the collapsing (the samples  are never exactly at the same position and the gradient can be very different) if it helps for the optimization problem as is well known from the Wasserstein GAN literature.

Due to the remaining concerns of the majority of reviewers, the AC recommends a reject but encourages the authors to resubmit the paper after taking into account the reviewers comments and the questions about the regularization .

---

> ### Author Response · Authors · 2021-02-03
> **Notes on the contrastive regularizer**
>
> Thank you for your feedback.
>
> However, we respectfully disagree with the following issues raised w.r.t. the contrastive regularizer, namely:
>
> 1)  "As a matter of fact looking at Figure 3 the regularization will spread the samples in the distributions, making the prototype more similar ." ------> While this might seem the case in 2 dimensions, it is likely not true in higher dimensions. In practice, the model has no incentive in making all prototypes similar, as that would mean a final vector of identical distances which is not discriminative enough compared to a vector of distinct numbers.
>
> 2) "The argument about the sample collapse is not good enough because if the sample collapse completely during the optimization the method converges toward prototype L2 which is not the case since the proposed approach is much better than L2 even without regularization. " ---> Yes, but the collapse is mostly local, i.e. some clusters of the point cloud collapse, not the entire point cloud. We argue that this is still a weakness compared to fully uncollapsed point clouds.
>
> 3) "Also the Wasserstsein can reverse the collapsing (the samples are never exactly at the same position and the gradient can be very different) if it helps for the optimization problem as is well known from the Wasserstein GAN literature." ----> We argue in appendix A.1 that this does not happen, i.e. it is a non-escaping situation when using gradient-based learning methods. The reason is that gradients of these collapsed points would become and remain identical, thus nothing will encourage them to "separate" in the future. This happens because we use a distance based loss in our model, as opposed to GAN models.